# Qualitative and Quantitative Assessment of Buckwheat Husks as a Material for Use in Therapeutic Mattresses

**DOI:** 10.3390/ijerph18041949

**Published:** 2021-02-17

**Authors:** Agnieszka Nawirska-Olszańska, Adam Figiel, Elżbieta Pląskowska, Jacek Twardowski, Elżbieta Gębarowska, Alicja Z. Kucharska, Anna Sokół-Łętowska, Radosław Spychaj, Krzysztof Lech, Marek Liszewski

**Affiliations:** 1Department of Fruit, Vegetable and Plant Nutraceutical Technology, Wrocław University of Environmental and Life Sciences, Chełmońskiego 37, 51-630 Wrocław, Poland; agnieszka.nawirska-olszanska@upwr.edu.pl (A.N.-O.); alicja.kucharska@upwr.edu.pl (A.Z.K.); anna.sokol-letowska@upwr.edu.pl (A.S.-Ł.); 2Institute of Agricultural Engineering, Wrocław University of Environmental and Life Sciences, Chełmońskiego 37, 51-630 Wrocław, Poland; adam.figiel@upwr.edu.pl (A.F.); krzysztof.lech@upwr.edu.pl (K.L.); 3Department of Plant Protection, Wrocław University of Environmental and Life Sciences, 24A Grunwaldzki Sq., 53-363 Wrocław, Poland; elzbieta.plaskowska@upwr.edu.pl (E.P.); elzbieta.gebarowska@upwr.edu.pl (E.G.); 4Department of Fermentation and Cereals Technology, Wrocław University of Environmental and Life Sciences, Chełmońskiego 37, 51-630 Wrocław, Poland; radoslaw.spychaj@upwr.edu.pl; 5Institute of Agroecology and Plant Production, Wrocław University of Environmental and Life Sciences, 24A Grunwaldzki Sq., 53-363 Wrocław, Poland; marek.liszewski@upwr.edu.pl

**Keywords:** buckwheat husk, prophylactic mattresses, natural products (ecological), pressure ulcer, microorganisms, pests

## Abstract

Buckwheat husks are used in many therapeutic products such as pillows, mattresses, seats, etc. This material is proposed by producers for example for discopathy, back pain and head vasomotor disorders. Our studies evaluated the impact of using cotton mattresses with buckwheat husk fillings on people’s health condition. The main research was carried out on the group of 60 people divided into 3 groups (1—people with skeletal system problems, 2—people spending a lot of time lying with the probability of pressure ulcer formation and 3—healthy people). In addition, different tests have been carried out on the possibility of colonization of mattresses by fungi, bacteria and arthropod pests, and rheological, chemical and flammability tests. The research material in the form of buckwheat husks was tested in a diverse way. All tests indicate high usefulness of husks for therapeutic activity. This material was contaminated with fungi, bacteria and pests at a very low level, related to the natural colonization of buckwheat nuts during harvest and storage. The quality of the husks was also confirmed in rheological, chemical and flammability studies. Finally, this has also been confirmed in surveys conducted on people with health problems. The analyses show that the buckwheat husk is an excellent material that can be used to fill prophylactic mattresses. This has been confirmed by the results of laboratory tests and opinions of respondents using mattresses filled with buckwheat husk.

## 1. Introduction

Buckwheat is a dicotyledonous grain crop plant belonging to the genus *Fagopyrum* in the family Polygonaceae [1]. It belongs to the group of cereal plants due to the similar chemical composition of seeds, their use and agrotechnics. The production of buckwheat in Poland is relatively small compared to the production of the main cereals, however, due to the buckwheat harvest, our country is one of the leaders of the world [2]. Buckwheat (*Fagopyrum esculentum* Moench) is a plant with multilateral use. First of all, it is grown for grain called kernels. Kernels are processed into groats, flour and used as an additive to many food products, e.g., pasta and biscuits [3,4]. Due to the gluten-free protein, buckwheat kernels are a good raw material for the production of functional food, and recommended especially for people suffering from celiac disease. Buckwheat protein extracts lower the level of LDL and VLDL cholesterol and prevent the development of colorectal neoplasms by limiting the proliferation of neoplastic cells [5]. Health benefits attributed to buckwheat include lowering plasma cholesterol, neuroprotection, anticancer, anti-inflammatory, antidiabetic and improvement in high blood pressure. In addition, it has been reported that buckwheat has prebiotic and antioxidant properties [6].

Buckwheat is also a valuable honey plant that allows bee colonies to obtain significant amounts of honey under favorable vegetation conditions. The high content of vitamin C, available iron and protein make buckwheat honey extremely valuable in the treatment of anemia or nervous disorders [7]. Buckwheat kernels contain nutrients such as proteins, saccharides, lipids, elements and non-nutritive: dietary fiber and a fraction of starch resistant to enzymatic hydrolysis. Is known for its high nutritional value and bioactive components, particularly the content of rutin and quercetin [1]. Content of the listed components is affected by the species, variety and environmental conditions [4].

In Japan, the variation in seed shape and husk color was investigated in 56 native buckwheat varieties. Analysis of variance revealed highly significant differences among the varieties in seed shape characteristics and husk colors. The 1000-seed weight ranged widely, from 9.48 to 15.22 g, among the varieties. The variance analysis showed very significant differences between the varieties in terms of seed shape and scale color [8]. In Poland, buckwheat kernels from the national register of varieties were also analyzed (Kora, Luba, Panda). Physical characteristics, 1000 seeds weight and the share of the seed coat in the kernel weight slightly differed between the varieties. The examined buckwheat kernels show similar characteristics, but the Kora variety is distinguished by the amount of neutral fatty acids with n-6 and n-9 unsaturated acids, the share of resistant starch and a soluble fraction of dietary fiber [4]. The buckwheat husks exhibited large differences between varieties in protein content (3.0–6.5%), bound phenolics (6.7–26.1 mg GA/g) and total phenolics content (32.4–58.6 mg GA/g) [9].

Buckwheat husk can also be used as a filling in medical mattresses. For the first time in the world, buckwheat husks have been used as a filling in pillows and sleeping mats in Japan [3]. Such pillows are traditionally used in Japan and Korea for the sake of health and comfort. Recently, their popularity has increased in Europe and North America, and buckwheat pillows are now regularly sold and advertised on TV [10]. Poland is the main exporter of buckwheat husks to Japan. Currently, there are also companies in our country that manufacture mattresses, pillows and seats filled with buckwheat material [3]. However, there are no scientific reports on products filled with buckwheat husks.

Therapeutic products from buckwheat husks (pillows, mattresses and seats) are recommended for discopathy, back pain and head vasomotor disorders. Buckwheat husk filling adapts to the position of the body, very quickly absorbs moisture and do not heat up. An important feature of these products, mainly due to the presence of tannins, rutin or quercetin is the inhibition of the development of harmful microorganisms, mites, insect pests, fungi and bacteria [3,11]. During the storage of buckwheat husks and the use of products made from them, it may come to be inhabited by polyphagous storage pests or the development of undesirable microorganisms. Pests eat and damage seeds and indirectly also have a negative effect by contaminating the seeds with dead individuals and feces. They also cause an increase in temperature and humidity, which creates favorable conditions for microbial growth. Insects are also carriers of pathogenic microorganisms that cause many diseases [12].

Buckwheat kernels are distinguished by biological, dietary and prophylactic and therapeutic properties, however phenolic compounds, tannins and their interactions with proteins may cause allergies [13]. Buckwheat allergy may occur in various situations, even in countries with a low average consumption of buckwheat dishes and low awareness of this type of allergy [14]. Allergy to buckwheat induces an IgE-mediated immediate-type I hypersensitivity reaction, which has been found after ingestion of buckwheat, occupational exposure to buckwheat or domestic exposure after sleeping on a pillow stuffed with buckwheat husks, however, studies did not reveal an isolated atopic occurrence [10]. To estimate the risk of buckwheat allergy, epidemiological studies are needed, especially in subgroups with high consumption of buckwheat food or the use of buckwheat husk pillows [14].

Throughout history, people have been looking for a comfortable place to rest. From the beginning of humanity many varieties of straw, feathers and later synthetic mattresses materials were created. Primary surfaces, mattresses and secondary surfaces evolved from natural products to foams and polyurethane. Currently, due to many years of research and changes in mattresses, medical problems, such as pressure ulcers, have been partially resolved, but they are still the subject of many observations and studies.

A very important issue in the treatment of bedridden patients—constantly lying is the prevention of pressure ulcers (PUs). This applies to both hospitals, in particular ICUs, and hospices and patients at home, and those people with restricted mobility. Many factors contribute to the increased susceptibility of patients to the development of PU: acute and chronic disease processes, immobilization, advanced age, high body mass index (BMI), sensory perception disorders, altered tissue perfusion and malnutrition. Such factors are often widespread, if they are not the norm in critically ill patients [15,16,17].

Treatment of pressure ulcers is problematic due to the presence of many comorbidities, chronic pressure ulcers in patients, and often also due to the relative lack of knowledge of the doctor in regard to treatment options. Some studies indicate that the main risk of pressure ulcers in hospitals occurs in intensive care (ICU) [18]. Many papers have been devoted to the prevention of pressure ulcers. Methods of local wound healing, methods of purification, diagnosis of infections, feeding support and mainly ways of preventing pressure ulcers by applying various types of mattresses were investigated [19,20,21]. What we sleep on affects the comfort of sleeping, and thus directly affects our well-being the next day. Too soft filling of the mattress in some cases causes back pain, uncomfortable head position may translate into migraine headaches, while too hard mattresses can cause muscle pain. Mattresses with bioactive filling can significantly improve sleeping comfort, which is important for everyone, especially for the elderly people who have trouble sleeping in general, and people with skeletal problems (backache pains).

The aim of the study was to assess the usefulness of buckwheat husks to fill health-related mattresses, taking into account its chemical–physical properties and susceptibility to colonization by mold fungi, bacteria and insect pests. Direct analysis of the suitability of products containing the tested filling material was also tested in the mattress user’s surveys. In the future, the possible allergenic effect of the buckwheat husks (matrix) should be investigated. It may depend on the chemical composition of the husks, which depends on the variety, environmental conditions of buckwheat cultivation and other factors.

## 2. Material and Methods

### 2.1. Buckwheat Husks as Material to Research

The buckwheat husks used for the tests come from the manufacturer of rehabilitation mattresses, a Polish family company “Manufacturer of Medical Devices GAMA Paweł Kaperczak” [22]. A large part of its buckwheat products is exported. Husks were obtained from buckwheat kernels of the Luba variety. Buckwheat crops, the recipient of which is the company, are located in various regions of Poland (Lubelskie, Wielkopolskie and Podlaskie voivodships). The husk of this variety is distinguished by a pyramidal shape with high elasticity and a high content of cellulose-lignin compounds. The production process ensures maximum cleaning of the husks on special screens. The same husk, free from contamination, can also be used for the production of functional food. The material from which health rehabilitation mattresses and other products are made is 100% cotton.

### 2.2. Air Permeability in Buckwheat Mattresses

To determine the degree of air permeability of selected products made of material filled with a buckwheat husks, an innovative device designed and made at the Institute of Agricultural Engineering of the Wroclaw University of Environmental and Life Sciences, Poland, was used (Figure 1).

The formulas from (1) to (4) allow one to define the parameters necessary to determine the degree of air permeability through a product made of material filled with buckwheat husks.
(1)$$pd=hd\cdot \rho_{H_{2}O}\cdot g$$
(2)$$pc=hc\cdot \rho_{H_{2}O}\cdot g^{\;}$$
(3)$$pd=\frac{\rho_{p}\cdot v^{2}}{2}\;$$
(4)$$v=\sqrt{\frac{2\cdot p_{d}}{p_{p}}}=\sqrt{\frac{2\cdot hd\cdot p_{h_{2}o}\cdot g}{p_{p}}}\;$$
where: *pd*—dynamic pressure; *hd*—height of dynamic pressure; *ρ_H_*_2*O*_—water density; *ρ_p_*—air density; g standard value of gravitational acceleration (9.80665 m s^−2^); *pc*—total pressure; *hc*—height of total pressure and *v*—air velocity

The higher the velocity of the air stream *v* and the ratio of *pd* to *pc*, the higher the permeability through the tested product resulting from the lower resistance of this product against the air stream produced by the fan. The load F simulated the compressive pressure exerted by the human body when using the product.

### 2.3. Desorption Properties of Buckwheat Husks Including Water Vapor Desorption Kinetics

The kinetics of sorption and desorption of water vapor showing the behavior of mattresses at different external conditions was determined in the process of moisturizing and drying the product filled with buckwheat husk. The moisturizing took place in an environmental chamber (WK111^340^ GmbH, Tuttlingen, Germany) at 40 °C and 95% humidity, while drying was carried out in the ambient air at 22 °C and 45% humidity. This drying process enabled determination the dependence of water activity on the moisture content of the material. The water activity was determined using the AquaLab DewPoint 4TE meter (Decagon Devices Inc., Pullman, WA, United States), and the moisture content *MC* was determined by drying in a convection dryer SLW 115 (POL-EKO Apparatus sp.j., Wodzisław Śląski, Poland) at 105 °C for 24 h. The value of *MC* was calculated accordingly to the Formula (5):(5)$$MC=\frac{m_{1}-m_{2}}{m_{2}}$$
where: *m*_1_—mass of the material before drying and *m*_2_—mass of the material before drying.

The mass of material before and after drying was measured using a laboratory balance (PS300.R2, Radwag, Poland).

### 2.4. Mass Contents of Particles of Different Size in Combination with Bulk and Tapped Density

The mass fractions x_i_ was calculated on the basis of Formula (6):(6)$$xi=\frac{mi}{mc}$$
where *m_i_* is the mass of the given fraction a *m_c_* total mass of the sample subjected to sieve analysis carried out using a set of sieves with dimensions: 0.5, 1.25, 3.15 and 4.0 mm (ISO 3310-1). The flat sieves of frame diameter 200 mm were placed one above the other in a mechanical shaker WSU (Dozamet, Poznan, Poland), which was working at the frequency 2 Hz and amplitude 40 mm. In turn, the density of the husk in a dry state with a moisture content of 9% in the bulk and shaken state (after 5-time shaking down) was determined in a measuring cylinder with a capacity of 1 L.

### 2.5. The Compression Work and the Plastic Deformation in Subsequent Cycles of the Modified TPA Test

The compression work and the plastic deformation in subsequent texture profile analysis (TPA) test cycles was determined basing on the double compression test known as texture profile analysis (TPA), which was modified by deforming the sample to achieve load 200 N at three runs of double compression. The test was carried out using an Instron 5566 strength testing machine (Instron, High Wycombe, UK) equipped with compression plate having a diameter of 120 mm. The load value was adjusted to the surface of the clamping plate to simulate the pressure exerted by the human body during the use of the product. Figure 2 shows a typical plot obtained during the first run of the modified TPA test. Total deformation obtained under assumed load was determined at the first compression (a,b). During the return movement of the plate, the force value was dropping to zero (b,c) and the starting position was reached without the contact with the material (c,d). The second and subsequent compressions enabled determination of plastic deformation (d,e), which occurred without the contact with the material and elastic deformation, which was accompanied by the compressive force arising to the assumed load F_max_ (e,f). It is impossible to distinguish the plastic deformation at the first compression. The values of compression works during the first and second deformation were calculated as the areas under the curves obtained during the first compression (a,b) and the second compression (e,f), respectively.

### 2.6. Husks Health Condition

#### 2.6.1. Microbiological Analysis of Buckwheat Husks Risings

The material for testing was buckwheat husks intended for filling medicinal mattresses. The research analyzes were carried out in accordance with PN-ISO 21527-2:2009 [23]. The total number of bacteria and fungi was determined by the plate method using the PCA medium (Plate Count Agar, BTL, Łódź, Poland) for isolation bacteria and medium with yeast extract for isolation fungi. The tests were carried out on 3 representative samples. The final result is presented in colony forming unit per 1 g of husk (cfu g^−1^).

The samples were also tested for the presence of bacteria pathogenic for humans. For isolation of *Staphylococcus aureus Staphylococcus* agar (Sigma-Aldrich, St. Louis, MO, USA) medium with mannitol (10 g L^−1^) and 0.2% phenol red (12.5 mL L^−1^) was used. Only mannitol-positive colonies with a yellow clear zone were counted. For isolation *Pseudomonas aeruginosa* was using King A medium (BTL, Łódź, Poland) with trimethoprim (Sigma-Aldrich, St. Louis, MI, USA) (inhibiting Gram-positive bacteria and the Enterobacteriaceae family). Colonies producing blue-green pigment and dye-free colonies were counted. For the isolation of bacteria from the Enterobacteriaceae family including *Escherichia coli*, MacConkey’s medium with lactose (BTL, Łódź, Poland) was used. Lactose-positive colonies with a pink glow around the colonies were counted on the substrate. In order to determine the presence of endospore forming *Clostridium perfringens* Wilson–Blair’s medium (BTL, Łódź, Poland) was used. The culture was made by the flooding method, and the sample was previously subjected to a thermal shock (90 °C, 20 min). Bacteria *C. perfringens* grow in the form of black colonies. The control consisted of reference strains from the Polish Collection of Microorganisms: *S. aureus* PCM 2054, *P. aeruginosa* PCM 2058 and *E. coli* PCM 2057. All plates were incubated at 37 °C for 48 h.

#### 2.6.2. Analysis of Populations of Buckwheat Husks by Fungi

For the visual assess the health status, 500 husks were collected. They were divided into two groups: healthy scales and those with mycelium coating or rotting due to the development of fungi and/or bacteria. To assess the degree of colonization of buckwheat husks by fungi, 200 scales were collected. Half of them were laid directly on the PDA (BTL, Łódź, Poland) substrate (surface microflora), and the other half before the liner was disinfected with 0.5% sodium hypochlorite for 20 s (deep microflora). The fungal spore size was measured under an optical microscope. A 40× magnification of the microscope was used, according to the standards used in mycology. For identifying fungi, the following monographs were used [24,25].

### 2.7. Possibility of Insect Pests Development

To exclude the presence of entomological contamination of the husks provided and an unroasted buckwheat seed, the material was directly visually analyzed. The husks and seeds were stored in 5 kg bags with cotton coatings. At random, 10 samples of 100 g each were collected and analyzed using a stereo microscope (13–56× magnification) to detect the presence of living or dead organisms or their residues.

In further research 6 species of pests commonly found in seed and food warehouses were used to assess the potential of entomological colonization, i.e., confused flour beetle (*Tribolium confusum* Du Val), flat bark beetle (*Cryptolestes ferruginesus* Stephens), drugstore beetle (*Stegobium paniceum* Linnaeus), khapra beetle (*Trogoderma granarium* Everts), lesser grain borer (*Rhyzopertha dominica* Fabricius) and the Indianmeal moth (*Plodia interpunctella* Hübner). Various numbers of females belonging to the mentioned species were used for analyses depending on their availability (from 3 to 40 specimens of each species). Pest breeding and laboratory tests were carried out at the Department of Plant Protection of The University of Environmental and Life Sciences in Wroclaw, Poland. Mother populations of each species were bred separately in plastic containers with a capacity of 1.2 L. The specific culture was carried out in two thermostats (double chamber thermostatic type: ST2/2 BASIC from POL-EKO Apparatus sp.j., Wodzisław Śląski, Poland) at 30 °C, with 50% or 80% humidity for selected species (*T. granarium*), depending on their requirements. For the breeding of each species in each combination of experiments, plastic containers with a diameter of 5.5 cm and a height of 7 cm with ventilation openings were used. Each container contained 5 g of buckwheat husks and 5 g of unroasted buckwheat seeds under control, followed by 48 h storage pests, *T. confusum*, *C. ferruginesus*, *S. paniceum, T. granarium*, *R. dominica* and *P. interpunctella*. In the case of all beetles, 40 adults were released. In the case of the Indianmeal moth, 5 moths were released. The experiment was carried out for a total of 46 days. Observations of insects, grains and husks were carried out cyclically at about weekly intervals and consisted in counting of individual developmental stages, i.e., eggs, larvae, pupae and adults. In the assessment of the development potential, the following factors were taken into account: the time of development of particular stages of the pest at a specific temperature and humidity, and the possibility of developing subsequent generations of the pest.

### 2.8. Antioxidant Activity and Polyphenols

Antioxidant activity was determined by DPPH (radical scavenging activity 1,1-diphenyl-2-picrylhydrazyl radical), ABTS (radical scavenging activity 2,2′-azinobis-(3-ethylbenzothiazoline-6-sulfonic acid) and FRAP (reduce the ferric ion) methods. The total polyphenol content was determined by the spectrophotometric method with the Folin–Ciocalteu reagent. All determinations were performed in triplicates using Shimadzu UV–Vis 2401 PC spectrophotometer (Tokyo, Japan) [26].

### 2.9. The Presence of Cellulose-Lignin Compounds

Determination of cellulose content and detergent-acid lignin in the husks was determined basing on the detergent method developed by Nawirska and Kwaśniewska [27].

### 2.10. Fire Resistance of the Material Filled

The fire resistance test of material filled with buckwheat husk was carried out according to the method based on an open fire test. It was made on a stand consisting of a tripod and an acetylene torch. Additional equipment was a thermal imaging camera that allows recording the temperature of the material in the flame exposure zone at a distance of 25 cm from the burner.

### 2.11. Survey Research of Mattress Users

To determine the usefulness of buckwheat mattresses for preventive and therapeutic purposes, the most often used in market research survey method was used. It involves the researcher’s written communication with the respondent. This form of communication has some basic advantages, but also limitations. It is accurate and permanent, but more difficult than oral communication (both for the researcher and the respondent) and takes more time than, for example, an interview. The measurement tool used in the survey method is always a questionnaire, filled in by the respondent. The information collected using the survey method enables both the diagnosis and the market forecast. The survey is a method of indirect measurement characterized by the fact that the questionnaire goes directly to the respondent who responds in writing or electronically to the questions contained therein [28].

Survey questions:−Please, define your first impression of using a buckwheat husk mattress;−Please, specify your health condition: (a) completely healthy, (b) a person with periodic skin problems, (c) a person with periodic bone problems (spine, joints), (d) a person with periodic muscle problems, (e) bedridden (person lying down, requiring care) and (f) others;−What mattress have you used so far? (a) Very hard, (b) moderately hard, (c) hard, (d) normal and (e) soft;−How do you assess the hardness of the buckwheat mattress? (a) Very hard, (b) moderately hard, (c) hard, (d) normal and (e) soft;−Sleep comfort: (a) very high, (c) medium, (d) angry and (e) very angry;−How do you assess the use? (a) Easy to clean (washing, airing), I. yes, II. no and III. I have no opinion and (b) were there any stains from the filling during use I. yes, II. No and III. I have no opinion;−What is the effect of using a mattress on the skin? (a) Big (bedsores decreased and skin appearance improved), (b) very slight and (c) none;−Whether you have an allergy or other skin changes after using the mattress? (a) yes, what … and (b) no;−Please, describe in a few words your own feelings about the use of a buckwheat mattress.

The survey involved 60 people and the respondents were divided into three groups: people with skeletal problems, people spending a lot of time lying down (probability of pressure ulcer formation) and healthy people. The age range of the subjects was from 2 to 84 years. The study involved ten children (on behalf of younger children, the questionnaire was filled by parents), group of people spending a lot of time lying down and 26 men and 24 women proportionally in all test groups. The mattresses were tested by the children of the Foundation for Children and the Hospice of the Bonifratrzy Fathers in Wrocław, Poland, and individual persons.

### 2.12. Statistical Analysis

Table Curve 2D v5.01 (Systat Software, San Jose, CA, USA) enabled fitting the mathematical equation to experimental points with the highest possible value of the determination coefficient R2 and the lowest value of fit standard error FSE. Standard deviations were estimated by means of Microsoft Excel.

## 3. Results and Discussion

### 3.1. Physical Properties of the Husks

The stated by the respondents breathability of mattresses with filling in the form of buckwheat husks was also confirmed by the results of tests regarding air permeability of these mattresses, which assumed that the air permeability of the tested product is determined by the speed of the air stream suppressed. It turned out that the mattresses filled with buckwheat husks had greater air permeability compared to a traditional mattress with a filling in the form of a polyester sponge. Thus, the higher air permeability of the buckwheat husks fillings, resulting from the higher air velocity value, has been confirmed by respondents who previously used mattresses with traditional fillings. It should be added here that the greater air permeability and thus better ability to drain water vapor, which is an important component of moist air is of particular importance for maintaining the user’s comfort in case of excessive sweating and provides the possibility of fast drying after uncontrolled wetting or intentionally performed washing.

People with high sweating paid noticed a significant reduction of this ailment, which translated into a significant increase in sleeping comfort, it is related not only to the high airiness of the mattresses, but also to the course of the sorption curve (Figure 3) indicating that the buckwheat husk filling is relatively fast moisturizing, absorbing water vapor from the environment, also from the area moistened due to the user’s sweating. On the other hand, the course of the desorption isotherm (Figure 4) shows that the decrease in moisture content at the range from 0.25 kg water kg fresh matter ^−1^ resulted in a significant decrease of water activity, which determines the potential for microorganisms to grow. Within this range water vapor was also quickly transferred into the room in which the mattress was used.

The desorption isotherm was described using sigmoid Equation (7) with high quality of fitting confirmed by high value of determination coefficient R^2^ = 0.9979 and low value of fit standard error FSE = 0.0186.
(7)$$a_{w}=\frac{0.9691}{1+e^{-\frac{MC-0.1227}{0.044}}}$$

Although the results of biological tests inform about the high impact of humidity on the health condition of buckwheat husks, creating external conditions favoring the maintenance of products with a filling in the form of buckwheat husk in a dry state, was enough to prevent the growth of bacteria and molds, as was confirmed in the conducted mycological and microbiological tests. In this context, periodic washing of mattresses was a microbiologically safe operation because the course of the desorption curve (Figure 3) that indicates the ability of the buckwheat husk to dry quickly in natural conditions.

Appreciated by the respondents feature of mattresses filled with buckwheat husk, which is a good fit to the body results from the fractional filling composition, where the highest mass fraction of x_2_ = 0.8628 characterized by the fraction with a dimension between 3.15 and 4 mm while the smallest, amounting to x_1_ = 0.0017, fraction with a dimension over 4 mm (Table 1).

Such dimensional proportions of the filling components favored the optimal deformation of the mattress under the influence of surface pressure, resulting from the user’s body pressure being perceived as a “good fit”. Undoubtedly, such a positive experience was also influenced by the possibility of changing the volume of the filling associated with a relatively large range of density changes determined in the bulk and shaken state, ranging from 169.88 ± 1.66 to 204.39 ± 3.14 kg·m^−3^. In practice, the density change of the filling occurs during the deformation of the mattress under the influence of the external load on the part of the user, which is associated with a reduction in the volume of filling due to the packing. The possibility of packing the filling in the optimal range favors fitting the body to a deformable mattress reflecting the body shape in the area of contact. The positive experience of the respondents verbalized by the possibility of “good body fit” was also confirmed by the results of the modified TPA test (texture profile analysis), which indicated a high deformation value at the first deformation corresponding to the first user contact with the product. Then the greatest value of the compression work of the product was made, which decreased in subsequent cycles (Figure 5).

The reduction of the elastic deformation (∆L_e_), found in the sequence of subsequent load cycles, which takes place at the expense of the increase in plastic deformation (∆L_pl_), shown in Figure 6, made it possible to predict that the user of the product filled with buckwheat husk will experience the greatest deformation of the product during the first contact with its surface. It has been noticed that the next change in load when sitting or lying down will be associated with lower deformation of the product, which is perceived as a reduction in susceptibility to deformation and may have a positive effect in relation to certain skeletal disorders. This supposition was confirmed by the respondents’ feelings that the product is extremely comfortable and after some time the pain of the neck and the back area subsided.

It was suggested that mattresses filled with buckwheat husks could be dedicated to users who for health reasons were advised to use mattresses with low susceptibility to deformation caused by the influence of lying related loads. It should be noted that all activities related to the change of the product’s location, e.g., when moving, adjusting to the bed contour, turning, etc., reduce the packing of the husks and thereby increase the elastic deformation during reloading, which promotes the “regeneration” of mechanical and physical properties affecting user experience.

### 3.2. Health Properties of Husks

Buckwheat seeds, when harvested and stored, can undergo microbial contamination, and the source of contamination can be soil, air and water. This reduces the possibility of using the raw material in industry, and in the case of buckwheat husks as a material for filling medical mattresses. There are no scientific publications on the health issues of husks used to fill mattresses. Research on the healthiness of grains shows that the bacteria colonizing the grain belong mainly to genera *Pseudomonas* (e.g., *P. fluorescens* and *P*. *herbicola*), *Micrococcus*, *Lactobacillus* and *Bacillus* (e.g., *B. subtilis* and *B. cereus*). However, the dominant bacterial flora (90%) are Gram-negative rods of the genus *Pseudomonas* [29,30]. The number of aerobic bacteria in 1 g of raw material can reach several million CFU. The number of bacterial microflora present on the cereal grain is estimated for 5 × 10^3^ up to 1.6 × 10^6^ cfu in 1 g of raw material [29]. Of the buckwheat husk rinsings tested, 14.6 × 10^5^ cfu in 1 g were isolated. Gram-negative bacteria, isolated on King A substrate, accounted for 9.66 × 10^5^ cfu in 1 g, while the number of Gram-positive bacteria, isolated on a substrate with mannitol was 1.33 × 10^2^ cfu in 1 g (Table 2).

Grain and buckwheat seeds may also include pathogenic flora, including bacteria of the genus *Salmonella*, or *Escherichia coli*. Intestinal bacteria appearing on the material indicate that the product is contaminated by birds or rodents and organic fertilizers [31]. Occasionally, aerobic Gram-positive bacilli may also be present on the grain (*B. cereus*) and anaerobic (*Clostridium perfringens* and *C. botulinum*) producing endospores [30]. In the buckwheat husk material tested no type of *E. coli* bacteria were found (MacConkey’s medium), *P. aeruginosa* (King A medium) or *C*. *perfringens* (Wilson–Blair’s medium). Occurrence or the possibility of maintaining pathogenic bacteria in medicinal mattresses, filled with buckwheat husks, could pose a threat to the health of their users. During the storage of cereal grains in the right conditions, when its moisture content is reduced, most bacteria die out, and their number falls below 1000 cells per 1 g. In contrast, at higher humidity they exhibit saccharolytic and proteolytic properties and bacteria of the genus *Bacillus* also heat resistance of endospores, which contributes to the deterioration of the quality of the raw material [32]. When the moisture content of the grain exceeds 15%, the bacteria’s place can be occupied by dynamically developing mold [29,30].

The visual assessment carried out shows that the husks filling the examined material did not show any changes in the form of mild contamination or bacterial decay. The filamentous fungi pose the greatest threat to the stored product due to the ability to produce mycotoxins. The number of filamentous fungi in 1 g of grain can reach up to several thousand colonies [29,30]. The number of fungi isolated from the buckwheat husks rinsings was on a similar level and amounted to 1.15 × 10^3^ cfu in 1 g. In healthy grain, the level of mold fungi is low (from 0 to 5 × 10^2^ cfu g^−1^), in the grain that went bad molds appear in very large quantities (from 300 to 500 × 10^6^ cfu g^−1^) [33].

Fungi can settle cereal grains during the growing season (so-called field fungi) or develop during the storage period (so-called stored fungi). The first group includes fungi belonging to the genera: *Alternaria*, *Botrytis, Cladosporium, Fusarium, Helminthosporium* and to the other *Aspergillus*, *Penicillium, Cheatomium, Rhizopus* and *Mucor*. Storage fungi first infect the germ, where the humidity is slightly higher, causing “blue eye”, and later in favorable conditions develop rapidly [29,30].

Mycological analysis (Table 3) shows that their surface was more infected by fungi than their internal tissues. The majority of non-disinfected husks (95%) were surface-settled by fungi, and disinfected husks were infected at 68%. In both types of husks, the dominant species was *Penicillium notatum,* which in 75% inhabited the surface of the husk (a non-disinfected husk), and in 64% the internal tissue of husks (a disinfected husks). The internal tissues inhabited the *Alternaria alternata* fungus infecting buckwheat in the growing season (23% of all colonies). Buckwheat husks were also more frequently inhabited by storage fungi, such as: *A. flavus, A. niger, P. notatum, P. urticae* and *P. vermiculatum* than those that infect plants during the growing season (*A. alternata*). Among this microflora, toxin-producing species, such as: *A. flavus* whether *P. urticae,* can be the most dangerous and can to produce aflatoxins or patulin. However, in the study material they did not account for more than 5%. No mold of the husks was observed, but the presence of numerous types of fungi from the *Penicillium* group indicates unfavorable storage conditions. Fungi of this type usually inhabit seeds with moisture content above 17% [34].

### 3.3. Possible Development of Pests on the Buckwheat Husks

Buckwheat husks intended for insect breeding were analyzed in terms of contamination of entomological material. Visually and using a binocular, 10 samples of husks and seeds weighing 100 g each were analyzed. In only three samples of husks, single mites were found (total of 14 individuals). In the buckwheat husks samples provided for analyses, there was a small presence of other entomological contaminants, such as single: egg casings, butterfly’s head, wing of a fly, hymenopteran pupae, ladybird and mites. Considering the total sample size, it can be assumed that such a small number of contaminants did not affect the quality of the husks. In seven samples no living organisms were found. In the case of buckwheat seeds, no pests or residues were recorded. It can therefore be concluded that the samples submitted for testing were almost free from entomological contamination and properly stored.

The assessment of the possibility of feeding and development of selected pests was carried out on the buckwheat husk material. In these studies an analysis of the development of six pest species listed in the methodology was carried out, and the results are presented in Table 4. It was shown that two out of six tested pest species used buckwheat husks, i.e., *Trogoderma granarium* and *Tribolium confusum*. In the case of a *T. granarium*, out of 40 adults, some lay eggs from which 12 larvae (30%) developed. In subsequent observations, carried out 25 days after the application of adults, 1 living adult and 18 dead individuals were found. In the last observations, after 46 days from the beginning of the study, 17 new generation larvae and 1 adult specimen were identified. In the case of *T. confusum*, 40 adult insects were also applied. After a week, the found specimens were removed and five larvae were observed, which gave 12.5% of the individuals that started their development on buckwheat husks. During the next observation, one living adult was found, and in the last period only four dead larvae, which meant that the development of the species on the husks was completed. In a similar percentage, the *Stegobium paniceum* (12.5% of larvae) started to develop on their husks. In subsequent periods, four and two adults of this pest were found respectively. For the remaining species of beetles, i.e., *Rhyzopertha dominica* and the *Cryptolestes ferruginesus*, in the first observations, only one larva of 40 adults (2.5%) was recorded, which meant that the development of both species was very weak. Out of the five *Plodia interpunctella*, one larva (20%) developed, but in the following dates three adults (16.02) and five larvae (9.03) were found. In the last observation period no adult specimens were found, which also means that the development of the pest on the buckwheat husks ended.

Of the insect pests, the largest damage to the storage of buckwheat grains is caused by various species of beetles and moths [35]. Products of buckwheat origin can be attractive for many pests, including commonly occurring in Poland as a flour beetle (*T. confusum*), rust red flour beetle (*Tribolium castaneum*), khapra beetle (*T. granarium*) and other. According to Filipek [36] the flour beetle should show higher fertility and shorter development time on buckwheat husks than on other cereal products (e.g., oatmeal). Kordan and Gabryś [37] prove that factors affecting population growth of *T. confusum* and the effectiveness of food consumption were different in barley and buckwheat. The best product for *T. confusum* was barley flour, but the size of the population and the efficiency of food consumption were higher in the case of crushed porridge compared to the whole barley. As research shows, *T. confusum* cannot eat the whole grain, because its mouth is not adapted to bite large and hard pieces of food. The studies of Kordan and Gabryś [37] shows that the best buckwheat product for *T. confusum* was whole, unprocessed groat, but the development of the insect was relatively slow on this product. The decrease in the content of nutrients and B vitamins as a result of removal of the embryo and aleurone layer during buckwheat breaking process was probably the main factor influencing the development *T. confusum* [38,39].

### 3.4. Biological Properties of the Husks

The high antioxidant activity of buckwheat demonstrated in biological studies, favoring microbiological safety during the use of mattresses, could have improved the health of the skin observed by ill persons. The presence of cellulose-lignin compounds and active compounds from the polyphenol group could also have an advantageous effect. The specific chemical composition of buckwheat husks definitely contributed to obtaining beneficial results of entomological research from the point of use of mattresses. Users did not indicate any signs of settlement of the products used by pests.

In the obtained buckwheat husks samples, the polyphenol content was 541.36 mgGE/100 g. The obtained results were comparable to the results given by Zduńczyk et al. [40]. In the work of other authors, 930 to over 2000 mg in 100 g were determined [40,41]. As a result of the determination of the composition of phenolic compounds by HPLC, it was found that the main polyphenolic fraction of buckwheat husks are flavonols. The sum of their contents was 121.35 mg/100 g as converted to quercetinone 3-rutinoside. Determination of the content of these compounds in the buckwheat husk is difficult and therefore the results obtained by many authors differ significantly, i.e., from 10 to over 400 mg/100 g [42,43].

The cellulose content in the buckwheat husk was 31.46%, the detergent-acid fiber 70.83% and the detergent-acid lignin 39.36%. In research Dziedzic et al. [39], the content of the above components was 35.55%, 67.15% and 31.60%, respectively, which confirms the results obtained in these studies.

### 3.5. Fire Resistance Test

The results of the fire resistance test of the material filled with buckwheat husks showed that after 90 s, the surface of the material appeared discolored until the temperature was reached 225 °C (Figure 7a), and ignition of the sample occurred only after 4 min at a temperature exceeding 360 °C (Figure 7b). It is noteworthy that after removing the fire source the sample immediately went out without any signs of smoldering. This proves that filling in the form of buckwheat husks is a slow-burning material and therefore safe in terms of fire hazard in contrast to other materials used as a filling of mattresses, which additionally emit dangerous gases during the process of rapid oxidation.

### 3.6. Opinion of Mattress Users

The first associations that arose for respondents associated with the use of a mattress filled with buckwheat husks are: comfort 60%, good sleep 30%, curative effect 20%, natural 20%, pain relief 20%, airy 20% and sleep comfort 10%. In the mattresses the buckwheat husks moves gently in the channels, so that they do not create constant pressure on one part of the body. In this way, pressure sores are naturally prevented, and at the same time gently massages the whole body. When assessing the hardness of the mattresses, the respondents stated that they are moderately hard mattresses (80%), 10% said that the mattresses are normal and hard.

Anti-bedsore mattresses do not cause inflammatory and allergic symptoms. They improve the well-being of people who test both ill and healthy. Persons with minor ailments wake up more rested, those with spinal problems, except that they get rested, they also noticed that they function better throughout the day, and the incidence and level of back pain significantly decreased. Due to the fact that the mattresses are hard and resistant to crushing, they do not lose their shape when lying down.

In the biggest advantages of the tested mattresses, the respondents of all examined groups indicated good sleep and comfort (50% each), then the respondents pointed to naturalness (30%) and breathability and good fit to the body shape (20% each). The majority of respondents (70%) stated that they were well rested, while 30% of respondents stated that it is difficult to say whether they sleep or not and what the reasons are. In the long-term group of respondents, the respondents also pointed to non-bed sores, healing of already formed and generally good effects on the skin.

The biggest disadvantage of the buckwheat mattress, which the respondents were paying attention to was its weight (50%), then instability (20%) and lack of aesthetics (10%), with 20% of the respondents seeing no faults.

To the question, “What effect does the use of the mattress have on the skin?”, people included in the group of people with bone and healthy problems answered: none (10%), hard to say (40%) and small (50%). However, in the group of people lying the answers were diametrically different: large (70%), small (20%) and hard to say (10%). Long-lying persons paid attention to the reduction of redness and pressure ulcers, and the appearance of new wrinkles and pressure ulcers. None of the test subjects observed allergy or any negative skin changes. People with sensitive skin noticed, however, visible improvement in the appearance of the skin (no redness and no roughness of the skin). In addition, people with great sweating drew attention to its significant reduction during sleep, which translated into a significant increase in sleeping comfort.

People testing mattresses, who have problems with neck pain and the area around the cross, have found that they are extremely comfortable, and after a certain time of use the pain of the neck and the area around the cross have significantly decreased.

## 4. Conclusions

The functional properties of mattresses filled with buckwheat husks appreciated by users were confirmed by the results of physical and mechanical tests. This regards maintaining the user’s comfort ensured by high air permeability, which counteracts the effects of excessive sweating. The high ability to absorb water vapor and release it to the environment, confirmed by sorption and desorption hysteresis of the buckwheat husk, additionally helps to maintain cleanliness, enabling frequent washing without exposure to microbial contamination, which was confirmed by the course of desorption isotherm revealing significant decrease of water activity below moisture content 25%. Another advantage of studied mattresses comprises a good fit to the body resulting from the possibility of changing the volume of the filling associated with a relatively large range of density changes determined in the bulk and shaken state, which results from the appropriate fractional filling composition characterized by a significant predominance of particles with a size from 3.15 to 4 mm. A desirable fitting the body to a deformable mattress was also confirmed by the results of modified TPA test, which indicated a high value of compression work at the first deformation corresponding to the first user contact with the product.

The microbiological, mycological and entomological assessment may be an important indicator of the quality of the husks. The examined husks were contaminated with fungi, bacteria and pests residues at a low level, related to the natural colonization of buckwheat nuts during harvest and storage. Keeping the husks in a dry state is quite enough to prevent the growth of bacteria, mold fungi or pests. Buckwheat husks intended for insect breeding were analyzed in terms of contamination of entomological material. Of the six species of pest tested, buckwheat husks developed best on buckwheat husks in which up to 30% of the individuals started their development. However, in subsequent observation dates, small numbers of progeny were found. In a much smaller percentage, the beetles *Trogoderma granarium* and *Tribolium confusum* developed on the husks. In the case of other insect species taken for analysis, limited development possibilities were found. The small number of pests did not affect the quality of the husks.

Filling preventive mattresses with buckwheat husks seems to be an excellent solution both in the case of use by people with slight health problems and by bedridden persons. The high sleeping comfort and high durability are undoubtedly contributing to their high assessment by respondents confirmed by laboratory tests. The study also revealed that mattresses filled with buckwheat husks exhibit low flammability.

Although the respondents did not indicate allergic reactions, it would be necessary to carry out tests whether buckwheat hull mattresses do not cause allergies.

## Figures and Tables

**Figure 1 ijerph-18-01949-f001:**
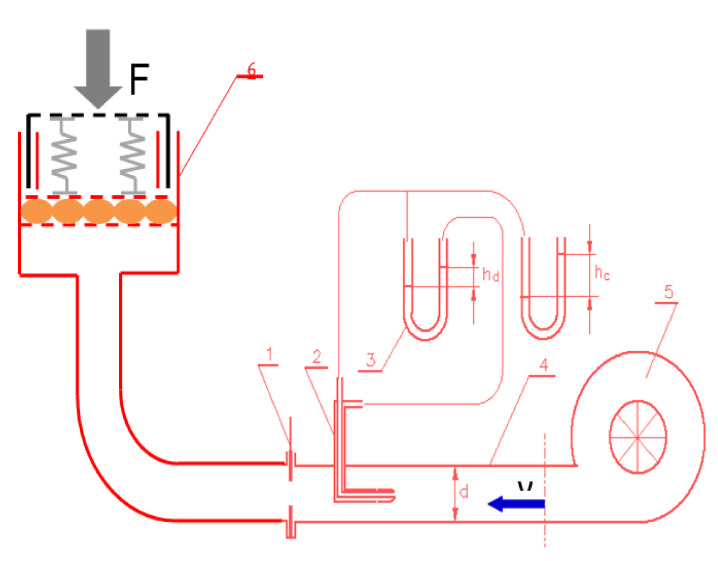
Schematic diagram of the device for determining the degree of air permeability through selected products made of a material filled with buckwheat hulls connected to an air flow generating set: 1—regulating valve, 2—Prandtl tube, 3—U-tube manometer, 4—air supply tube, 5—fan, 6—device for determining the degree of air permeability.

**Figure 2 ijerph-18-01949-f002:**
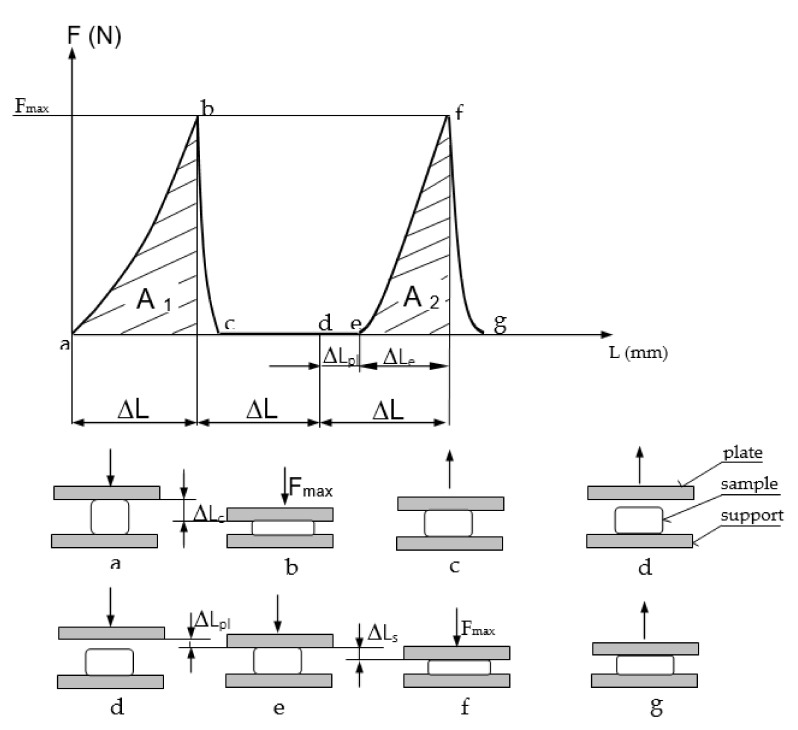
Successive phases in the first run of the modified texture profile analysis (TPA) test involving double compression of the sample to achieve the assumed load. F—compressive force, F_max_—maximum value of the compressive force, L—distance, ΔL—total deformation, ΔL_e_—elastic deformation, ΔL_pl_—plastic deformation, A1—work in the first compression cycle and A2—work in the second compression cycle.

**Figure 3 ijerph-18-01949-f003:**
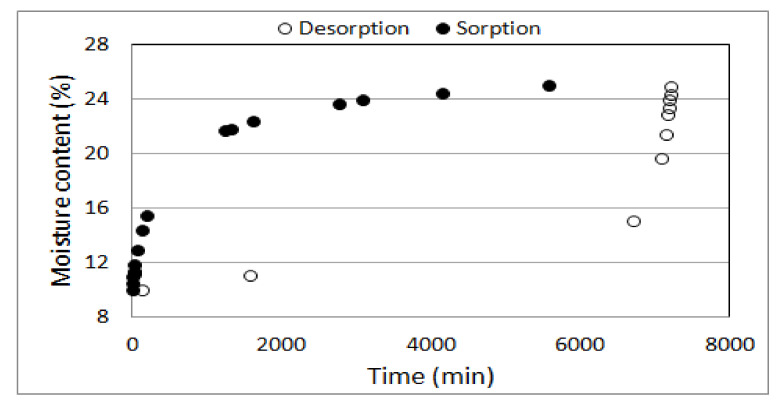
Example of sorption and desorption hysteresis of the buckwheat husks.

**Figure 4 ijerph-18-01949-f004:**
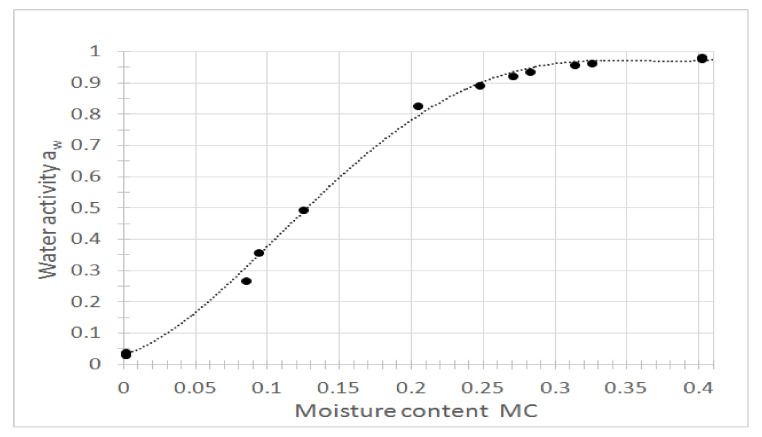
Desorption isotherm representing the relationship between moisture content and water activity of the buckwheat husks.

**Figure 5 ijerph-18-01949-f005:**
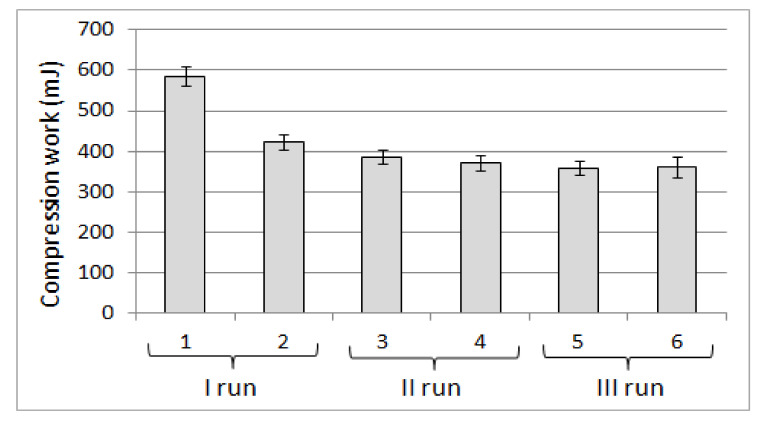
Compression energy in subsequent cycles of the modified TPA test carried out on a material filled with buckwheat husk with a moisture content of 9%.

**Figure 6 ijerph-18-01949-f006:**
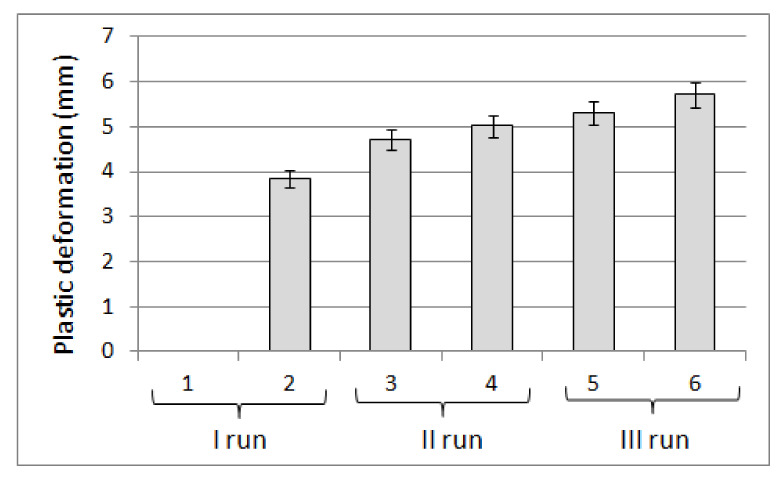
Plastic deformation in subsequent cycles of the modified TPA test carried out on a material filled with buckwheat husk with a moisture content of 9%.

**Figure 7 ijerph-18-01949-f007:**
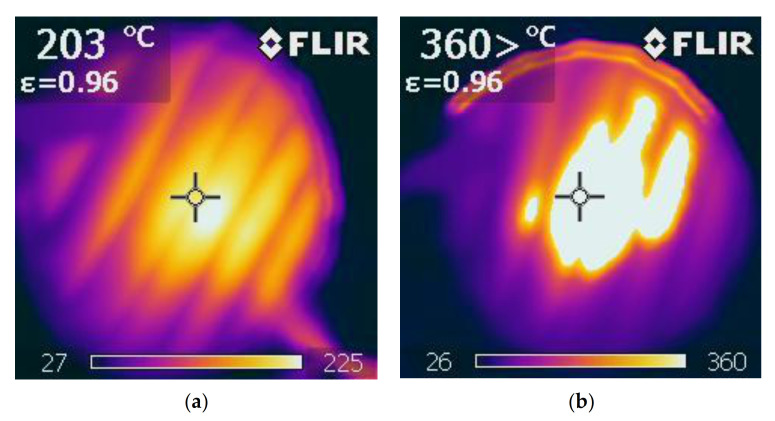
Infrared camera images of material filled with buckwheat husks subjected to a fire test: (**a**) no permanent color changes and (**b**) ignition of the sample.

**Table 1 ijerph-18-01949-t001:** Results of the sieve analysis of the buckwheat husk with mass fractions xi.

Dimension (mm)	Mass Fraction x_i_	Value	Standard Deviation
over 4	x_1_	0.0017	0.0001
3.15–4	x_2_	0.8628	0.0543
2–3.15	x_3_	0.0057	0.0006
1.25–2	x_4_	0.0390	0.0041
0.5–1.25	x_5_	0.0739	0.0052
below 0.5	x_6_	0.0169	0.0011

**Table 2 ijerph-18-01949-t002:** Total number of microorganisms isolated from the washings of buckwheat husks on different types of medium.

Microorganisms/Medium	cfu g^−1^ Buckwheat Husk
Total number of filamentous fungi and yeast on medium with yeast extract	1.15 × 10^3^ ± 0.12
Total number of bacteria on the PCA medium	14.6 × 10^5^ ± 1.15
Total number of bacteria on the *Staphylococcus* agar medium (Mannitolo-positive bacteria)	1.33 × 10^2^ ± 0.57 (not found)
Black colony bacteria on the Wilson–Blair medium (*Clostridium perfringens*)	0 (not found)
Total number of lactoso-positive bacteria (Gram-negative rods from the Enterobacteriaceae family) on the MacConkey medium (Gram-negative rods from the Enterobacteriaceae family)	0 (not found)
Gram-negative rods on King A medium (including *Pseudomonas aeruginosa* with green pigment)	9.66 × 10^5^ ± 1.52 (not found)

cfu—colony forming units for 1 g buckwheat husk, ± standard deviation of 3 repetitions.

**Table 3 ijerph-18-01949-t003:** Species of filamentous fungi isolated from of buckwheat husks.

No.	Species of Fungi	Husks		Husks	
		Non-Disinfected		Disinfected	
		∑	%	∑	%
1	*Alternaria alternata*	7	2	48	23
2	*Aspergillus flavus*	16	5	6	3
3	*Aspergillus niger*	16	5	2	1
4	*Mucor globosus*	3	1	15	7
5	*Mucor hiemalis*	6	2	-	-
6	*Penicillium notatum*	241	75	132	64
7	*Penicillium urticae*	-	-	2	1
8	*Penicillium vermiculatum*	3	1	-	-
9	*Rhizopus stolonifer*	29	9	2	1
The sum of isolated fungi		321	100	207	100
	Shell not grown with fungi (quantity pcs/300 pcs)	15 (5%)		96 (32%)	

**Table 4 ijerph-18-01949-t004:** Analysis of the development of storage pests on the buckwheat husk.

Species	Humidity %	23.01.	30.01		16.02		9.03		% of Individuals Starting Development
		No. Individuals Applied	Removal of Individuals	Larvae	Adults	Molt	Larvae	Adults	
*Trogoderma granarium*	80	40	32	12	1 (18 dead)	48	17	1	30
*Tribolium confusum*	50	40	6 (34 dead)	5	1 (3 dead)	18	4 dead	-	12.5
*Stegobium paniceum*	50	40	11 (29 dead)	5	4 (12 dead)	-	2 dead	2 (4 dead)	12.5
*Rhyzopertha dominica*	50	40	19 (6 dead)	1	9	-	1	-	2.5
*Cryptolestes ferruginesus*	50	40	11 (9 dead)	1	10	-	2 dead	-	2.5
*Plodia interpunctella*	50	5	5	1	3	-	5	-	20

## Data Availability

The data presented in this study are available on request from the corresponding author. The data are not publicly available due to privacy concerns.

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
