# Peer review of "Qualitative and Quantitative Assessment of Buckwheat Husks as a Material for Use in Therapeutic Mattresses"

_ijerph, 2021, doi:10.3390/ijerph18041949_

Round 1
Reviewer 1 Report
General comments
The manuscript presented for review covers a rather interesting topic, the use of buckwheat hulls as fillers for medicinal mattresses. Despite the fact that the topic is quite well known, many people still use this type of product insufficiently. Rarely preventively. The topic presenting the properties of buckwheat hulls is current. The article therefore has potential.
However, the manuscript requires a thorough revision, which should cover almost the entire manuscript: introduction, methodology, research results. The article has many technical and substantive shortcomings. Only examples: there is no detailed description of the research methodology, no statistical analysis, incomplete presentation of research results, no discussion of research in the case of physical research, etc. The research is extensive but requires refinement and ordering. Moreover, in my opinion, there should be a separate subsection describing the results of the surveys. Comparing survey results to empirical data is unclear (confusing).
Detailed comments below:
Line 39: The introduction is a cursory . It does not fully describe the topic of the article. The topic of the thesis is qualitative assessment ……, while the content contains very little information about the research of other authors in this area. Please extend the introduction to include a description of exemplary physicochemical properties of buckwheat hulls (refer to the literature).
Line 40: Add literature on the potential use of buckwheat.
Line 43: Add literature that confirms the healing properties of buckwheat
Line 59-61: If those mattresses are subject to many observations, give examples (refer to the literature)
Line 88: The methodology is described very briefly. Much important information is missing. It is not possible to replicate the test results by other researchers.
Line 88: The methodology does not contain a detailed description of the material to be tested (buckwheat hulls and the mattress itself). Add origin, material. Add as much information as possible about it.
Line 90: Based on this information, the research cannot be repeated.
Line 90 - 92: Please describe exactly how the test was performed. You can add a device schematic for the permeation test. On the diagram, indicate the location of the sensors and the direction of air flow.
Line 105: On what basis was this fact found in the research methodology?
Line 108: Add more information about the equipment (manufacturer, country, city)
Line 109: Was it a convection dryer? Report how the moisture content of the product was measured.
Line 114: Specify the sieve analysis device (vibration amplitude, frequency Hz). What kind of sieves were these. Give the standard of sieves.
Line 118: There is no test equipment, no standard, no relevant information about this test. There is also no standard on the basis of which the tests were performed. This should be added.
Statistics analysis should be added to the tests of physical properties. At least error whisker or standard deviation.
Line 129: Provide more information about the substrate, (manufacturer, origin of the substrate)
Line 155: Exactly explain the term "diagnostic keys". There is no information about the microscope used, magnifications, etc.
Line 161: There is no information about the microscope used, the objective magnification used, etc.
Line 173: Add more details about the equipment (manufacturer, country, city)
Line 188: Add an extension to these acronyms
Line 189: Spectrophotometric analysis device model (manufacturer, country, city)
Line 193: Were exactly the same methods and apparatus used?
Line 211: Describe what questions were in the survey questionnaire.
Line 222: There is no reference to the results obtained in the description of the physical properties. There are no visualized results that can be referred to (e.g. result tables)? In addition, the results of the physical properties should be statistically processed. At least error or SD whiskers should be used.
There is also no discussion of the results, the studies are not compared with the studies of other authors. This should be added.
Line 235: That was the research that was referenced to the survey?
Line 240: How do you know that.? Missing an isotherm graph reference?
Line 241: Si layout.
Line 260: Value should be in MPa
Line 271: In the graph, add the error whiskers. Describe the legend more precisely.
Line 273: In the methodology, explain what so this is delta Ls.
Line 325: Since the assessment was visual, some kind of microscope photo may be presented, if any.
Line: 434-435: Should be Figure 2.
Line 489: Are these mattresses really fireproof?
Line 472: Requests should include more information about the physics experiments.
Author Response
Answer to Reviewers Comments
General Answer
The work has been corrected precisely according to the reviewers' suggestions. They are given as detailed as possible. A large part has been added to the introduction and methods. Where possible, new references are also given. Below are detailed references to the comments of both reviewers. We hope that both the corrections in the text (highlighted in red) and the answers below are presented clearly enough.
Sincerely,
Authors
Rec 1
The manuscript presented for review covers a rather interesting topic, the use of buckwheat hulls as fillers for medicinal mattresses. Despite the fact that the topic is quite well known, many people still use this type of product insufficiently. Rarely preventively. The topic presenting the properties of buckwheat hulls is current. The article therefore has potential.
However, the manuscript requires a thorough revision, which should cover almost the entire manuscript: introduction, methodology, research results. The article has many technical and substantive shortcomings. Only examples: there is no detailed description of the research methodology, no statistical analysis, incomplete presentation of research results, no discussion of research in the case of physical research, etc. The research is extensive but requires refinement and ordering. Moreover, in my opinion, there should be a separate subsection describing the results of the surveys. Comparing survey results to empirical data is unclear (confusing).
Detailed comments below:
Line 39: The introduction is a cursory . It does not fully describe the topic of the article. The topic of the thesis is qualitative assessment ……, while the content contains very little information about the research of other authors in this area. Please extend the introduction to include a description of exemplary physicochemical properties of buckwheat hulls (refer to the literature).
Line 40: Add literature on the potential use of buckwheat.
Line 43: Add literature that confirms the healing properties of buckwheat
Line 59-61: If those mattresses are subject to many observations, give examples (refer to the literature)
Answer to the above four comments
All the above suggestions were taken into account and Introduction section was substantially completed
Line 88: The methodology is described very briefly. Much important information is missing. It is not possible to replicate the test results by other researchers.
Line 88: The methodology does not contain a detailed description of the material to be tested (buckwheat hulls and the mattress itself). Add origin, material. Add as much information as possible about it.
Line 90: Based on this information, the research cannot be repeated.
Line 90 - 92: Please describe exactly how the test was performed. You can add a device schematic for the permeation test. On the diagram, indicate the location of the sensors and the direction of air flow.
Answer to the above four methodological comments
All the above comments from the reviewers were included in the revised version of the manuscript. The methodology has been supplemented in detail wherever possible. Another subsection 2.1. was added to complete origin of buckwheat for research. The schematic diagram of the device and relevant description was provided.
Line 105: On what basis was this fact found in the research methodology? on line 105 it is gravitational acceleration - I think this is given generally. This value was adopted by the 3rd General Conference of Weights and Measures in 1901
Answer: The term “gravitational acceleration” was replaced with the term “standard value of gravitational acceleration (9.80665 m s-2)”.
Line 108: Add more information about the equipment (manufacturer, country, city)
Answer: The missing information about the equipment was supplemented.
Line 109: Was it a convection dryer? Report how the moisture content of the product was measured.
Answer: The type of convection dryer and the method of moisture content determination were included.
Line 114: Specify the sieve analysis device (vibration amplitude, frequency Hz). What kind of sieves were these. Give the standard of sieves.
Answer: The information regarding sieve analysis was supplemented.
Line 118: There is no test equipment, no standard, no relevant information about this test. There is also no standard on the basis of which the tests were performed. This should be added.
Statistics analysis should be added to the tests of physical properties. At least error whisker or standard deviation.
Answer: The missing information was provided. The standard deviations were added to the mean values.
Line 129: Provide more information about the substrate, (manufacturer, origin of the substrate)
Answer:
Information about the substrate (manufacturer, origin of the substrate) was completed at a point 2.6. Husks Health Condition
Line 155: Exactly explain the term "diagnostic keys". There is no information about the microscope used, magnifications, etc.
Answer:
Information about "diagnostic keys" and the microscope used, magnifications etc. was done at a point 2.6.2. Analysis of populations of buckwheat husks by fungi.
Line 161: There is no information about the microscope used, the objective magnification used, etc.
Answer: has been added
Line 173: Add more details about the equipment (manufacturer, country, city)
Line 188: Add an extension to these acronyms
Answer: has been added
Line 189: Spectrophotometric analysis device model (manufacturer, country, city)
Answer: has been added
Line 193: Were exactly the same methods and apparatus used?
Answer: Yes
Line 211: Describe what questions were in the survey questionnaire.
Answer: has been added
Line 222: There is no reference to the results obtained in the description of the physical properties. There are no visualized results that can be referred to (e.g. result tables)? In addition, the results of the physical properties should be statistically processed. At least error or SD whiskers should be used.
Answer: The results were presented in the form of figures and table with appropriate statistical analysis.
There is also no discussion of the results, the studies are not compared with the studies of other authors. This should be added.
Answer: There is no such research, discussion was implemented where possible
Line 235: That was the research that was referenced to the survey?
Answer: yes
Line 240: How do you know that.? Missing an isotherm graph reference?
Answer: The missing isotherm graph was provided to the manuscript.
Line 241: Si layout.
Answer: The mistake regarding moisture content and water activity was corrected
Line 260: Value should be in MPa
Answer: The value concerns density and therefore the unit is correct. The action of the external load on the surface of the mattress, which is related to pressure was only mentioned, not determined.
Line 271: In the graph, add the error whiskers. Describe the legend more precisely.
Answer: The graph was simplified to be in line with the text of the manuscript and the description was improved.
Line 273: In the methodology, explain what so this is delta Ls.
Answer: The abbreviation delta Ls was replaced with more appropriate delta Le which was described in the methodology.
Line 325: Since the assessment was visual, some kind of microscope photo may be presented, if any.
Answer:
Because no fungal bloom and bacterial slime/mucus were observed on the scales under a stereoscopic microscope, photos of healthy buckwheat husks were not taken.
Line: 434-435: Should be Figure 2.
Answer: The numbering of figures has been improved.
Line 489: Are these mattresses really fireproof?
Answer: The term “fire-resistant” was replaced with the correct term “low flammability”
Line 472: Requests should include more information about the physics experiments.
Answer: Relevant information about the physics experiments was included to the Conclusions
All considered, the article is well written, the presented experiments and their discussions are clearly explained and a revision of the aforementioned points could greatly improve the impact of this work.
Of less importance, the formatting of the formulas, i.e. lined 97-100, seems to be off, and the fonts in the abstract (lines 19, 27-28 and 30-32) and in the text (i.e. 65-69) should be uniformed with the whole manuscripts.
Answer:
It was corrected, but it was probably a formatting or publisher's error.

Reviewer 2 Report
The article by Nawirska-Olszańska et al. concerns with the assessment of buckwheat husks as material for use in therapeutic mattresses. The topic is interesting, but the I feel the experimental implant to be lacking.
In particular there is no comparison between "legacy" mattresses and the ones based on buckwheat husks. Moreover, no attention was given to the topic of potential allergies that this matix could affect.
The article concerns the physical characterization of mattresses made up from buckwheat husks, as well as an analysis of the potential pathogens that could grow in such ambients. The characterization is well conducted and the results are interesting, but there is no comparison (of the same parameters) with mattresses actually employed. As such, how could this paper be a "quantitative" or "qualitative" assessment of the use of buckwheat husks as a materials?
As such, a set of measures on actually employed materials should be carried out, or at least a comparative literature study that highlight how the measured parameters for the buckwheat husk mattresses are well within the accepted condition for this use. This comparison is of great importance since the formulation of a novel medical device requires an improvement in either the performance or efficiency.
Moreover, there was no mention of the necessity to conduct tests in order to assess the potential of buckwheat husks to induce allergic reactions in patients. This one is also an important parameter to evaluate the suitability of a material in such fields. I understand that not all laboratories are equipped for such analysis, but this should be pointed out in the article conclusion as a further study to be carried out.
All considered, the article is well written, the presented experiments and their discussions are clearly explained and a revision of the aforementioned points could greatly improve the impact of this work.
Of less importance, the formatting of the formulas, i.e. lined 97-100, seems to be off, and the fonts in the abstract (lines 19, 27-28 and 30-32) and in the text (i.e. 65-69) should be uniformed with the whole manuscripts.
Author Response
General Answer
The work has been corrected precisely according to the reviewers' suggestions. They are given as detailed as possible. A large part has been added to the introduction and methods. Where possible, new references are also given. Below are detailed references to the comments of both reviewers. We hope that both the corrections in the text (highlighted in red) and the answers below are presented clearly enough.
Sincerely,
Authors
Rec 2
The article by Nawirska-Olszańska et al. concerns with the assessment of buckwheat husks as material for use in therapeutic mattresses. The topic is interesting, but the I feel the experimental implant to be lacking.
In particular there is no comparison between "legacy" mattresses and the ones based on buckwheat husks. Moreover, no attention was given to the topic of potential allergies that this matix could affect.
The article concerns the physical characterization of mattresses made up from buckwheat husks, as well as an analysis of the potential pathogens that could grow in such ambients. The characterization is well conducted and the results are interesting, but there is no comparison (of the same parameters) with mattresses actually employed. As such, how could this paper be a "quantitative" or "qualitative" assessment of the use of buckwheat husks as a materials?
As such, a set of measures on actually employed materials should be carried out, or at least a comparative literature study that highlight how the measured parameters for the buckwheat husk mattresses are well within the accepted condition for this use. This comparison is of great importance since the formulation of a novel medical device requires an improvement in either the performance or efficiency.
Answer:
It was not possible to conduct research on used mattresses because they were donated to respondents and are still in use. We agree, it should be more interesting take into account the assessment of the quality of the material already used, but it should be noted that the analysis carried out by us gives a lot of different information about the material used to make medicinal products, it is a very broad assessment carried out by specialists from at least a few fields and all the results obtained indicate high quality buckwheat husks.
Moreover, there was no mention of the necessity to conduct tests in order to assess the potential of buckwheat husks to induce allergic reactions in patients. This one is also an important parameter to evaluate the suitability of a material in such fields. I understand that not all laboratories are equipped for such analysis, but this should be pointed out in the article conclusion as a further study to be carried out.
Answer:
Although the respondents did not indicate allergic reactions, it would be necessary to carry out tests whether buckwheat husk mattresses do not cause allergies.

Round 2
Reviewer 1 Report
The authors made most of the corrections included in the review. The introduced corrections are at a satisfactory level.
Reviewer 2 Report
Considering the impossibility to carry out a further comparison with "legacy" pillows, the article can be published in the current form.